# Many foliar endophytic fungi of *Quercus gambelii* are capable of psychrotolerant saprotrophic growth

Emily Weatherhead, Emily Lorine Davis, Roger T. Koide[ID]*

Department of Biology, Brigham Young University, Provo, UT, United States of America

* rogerkoide@byu.edu

**Data Availability Statement:** All relevant data are within the paper and its Supporting Information files.

## Abstract

Many endophytic fungi have the potential to function as saprotrophs when living host tissues senesce and enter the litter pool. The consumption of plant litter by fungi obviously requires moisture but, in the arid, western USA, the native range of *Quercus gambelii* Nutt., most of the precipitation occurs during the coldest months of the year. Therefore, we hypothesized that the endophytic fungi of *Q. gambelii* have the potential to function as psychrotolerant saprotrophs, which we defined in this study as an organism capable of significant growth on leaf litter at 5˚C. We further hypothesized that a tradeoff exists between growth of endophytic fungi at 5˚C and at 17˚C such that fungal isolates are either cold- or warm-temperature specialists. Consistent with our first hypothesis, we found that 36 of our 40 isolates consumed leaf litter at 5˚C, but there was a surprisingly high degree of variability among isolates in this ability, even among isolates of a given species. Contrary to our second hypothesis, there was no tradeoff between saprotrophic growth at 5˚C and saprotrophic growth at 17˚C. Indeed, the isolates that grew poorly as saprotrophs at 5˚C were generally those that grew poorly as saprotrophs at 17˚C. By virtue of being endophytic, endophytic fungi have priority in litter over decomposer fungi that colonize plant tissues only after they enter the litter pool. Moreover, by virtue of being psychrotolerant, some endophytic fungi may function as saprotrophs during the cold months of the year when moisture is temporarily available. Therefore, we suggest that some endophytic fungi of *Q. gambelii* could play significant ecosystem roles in litter decomposition and nutrient cycling.

## Introduction

In arid ecosystems, microbe-mediated litter decomposition and nutrient cycling can be limited by moisture [1], but this may depend on the time of year as some arid ecosystems are only seasonally dry. In the Great Basin desert of the western United States, for example, inadequate moisture frequently limits microbial decomposition of litter in the warmer months of the year, but it may not limit decomposition in the winter and early spring when evaporation is low and precipitation is relatively high [2]. Because moisture is least limiting in the winter and early spring, microbial decomposition of litter in the Great Basin of the western United States

**Funding:** The authors acknowledge financial support from the Roger Sant Foundation and the Department of Biology, Brigham Young University. The funders had no role in study design, data collection and analysis, decision to publish, or preparation of the manuscript.

**Competing interests:** The authors have declared that no competing interests exist.

requires that some saprotrophic microorganisms have significant metabolic activity under cold conditions [3,4]. Such microorganisms are referred to as psychrotolerant, psychrotrophic or psychrophilic [5,6]. We hereafter refer to this phenomenon as psychrotolerance.

Endophytic fungi very commonly colonize living plant tissues [7]. In some cases the host plant benefits from their colonization, which may improve seed germination [8], vegetative growth [9] and tolerance to stresses such as drought [10,11], extreme temperature [12,13], salinity [13,14], heavy metals [15] and herbivory [16,17]. Moreover, upon senescence of the host tissues they occupy, many endophytic fungi are capable of obtaining nutrition saprotrophically [18–23]. Indeed, by colonizing living plant tissues, endophytic fungi are essentially pre-colonizing litter. Therefore, endophytic fungi are among the first with access to litter as a source of energy and nutrients [21,24]. Priority access to a resource frequently results in a competitive advantage [25–27]. Thus, in litter, endophytic fungi may have a competitive advantage over decomposer fungi that colonize the plant tissues only after they senesce [28].

If an endophytic fungus were strictly biotrophic, it would only have to maintain activity during the warmer growing season. However, if an endophytic fungus were to function as a saprotroph in the Great Basin, where moisture is least limiting in the winter and early spring, it would be advantageous to be psychrotolerant. Our first hypothesis, then, is that endophytic fungi isolated from leaves of *Quercus gambelii* Nutt. are psychrotolerant saprotrophs, and that there are significant differences among species in their ability to function as psychrotolerant saprotrophs. Organisms frequently exhibit functional tradeoffs. For example, tradeoffs have been observed in fungi between growth and enzymatic activity (Zheng et al., 2020), between growth and reproduction [29,30] and between growth and defense [31]. Still other tradeoffs occur because organisms frequently cannot excel along the entire length of an environmental gradient. For example, plants that are active in cool climates may adapt their physiology to lower temperatures such that they do not perform well at higher temperatures, while plants that are active in warm climates adapt their physiology to higher temperatures [32]. Therefore, our second hypothesis is that there is a tradeoff among the endophytic fungi isolated from *Q. gambelii* such that they specialize at either winter or summer temperatures.

## Materials and methods

### Study sites and leaf sampling

*Q. gambelii* is a small tree or shrub found mainly in Utah, Arizona, New Mexico and Colorado. In Utah it is widely distributed between approximately 1,200 and 2,400 meters in elevation [33]. In order to sample leaves from across the elevational range of the species in Utah, whole, unblemished leaves were collected on 24 September 2019 at Devil's Kitchen (39.48'12.27"N, 111.41'17.96"W), elevation 2,553 meters, near Payson, Utah and on 8 October 2019 in Slate Canyon (40.13'30.59"N, 111.37'21.9"W), elevation 1,553 meters, in Provo, Utah. Air temperatures were monitored at the Devil's Kitchen site from 24 Jun 2019 to 24 Sep 2019 using a temperature data logger (HOBO UA-001-64, Onset Computer Corporation, Bourne, MA, USA), set to log hourly. At Devil's Kitchen, 15 leaves were collected from each of 7 trees and, in Slate Canyon, 10 leaves were collected from each of 6 trees, totaling 165 leaves. No sampling permits were required because of the small number of samples taken. In the field, leaves were stored in plastic bags and placed in a cooler on ice. Later on the same day, the leaves were transferred to an incubator at 5°C in the laboratory.

### Fungal isolation

We isolated fungi from *Q. gambelii* leaves in a laminar flow hood to insure sterility. External leaf surfaces were sterilized by submerging leaves in 70% ethanol for 2 seconds, immediately

submerging them in 3% sodium hypochlorite for 2 minutes, similar to Arnold et al. [7], rinsing away the hypochlorite by submerging leaves sequentially in three beakers of sterile water, then submerging, again, in 70% ethanol, after which they were laid to dry in a sterile petri dish. Each surface-sterilized leaf was subsampled once using a sterilized, 6 mm diameter, paper hole punch, taking care to re-sterilize the punch between leaves from individual trees with ethanol that was flamed off. Each leaf disk was placed on 2% malt extract agar in its own petri dish, sealed with parafilm and incubated initially at 10˚C, a temperature that had previously been used to isolate psychrotolerant endophytic fungi [34]. After 3 weeks, fewer than 10 total fungal colonies grew out from the entire collection of leaf disks. These were isolated to their own dishes. The original dishes containing the leaf disks were then transferred to a 17˚C incubator for an additional 3 weeks. Many new fungi growing from the leaf disks at 17˚C were thus isolated. When there were multiple fungal colonies growing from a single leaf disk, each was isolated separately.

## Sequencing and assigning taxonomy of fungal isolates

A total of six hundred seventy-five fungal isolates were sorted into morphological groups based on color, hyphal growth pattern and mycelial density. From these groups, 40 isolates were chosen for study, each of which was sequenced. Each isolate was subject to direct PCR [35–37] of the ITS region using APEX 2 Hotstart Master Mix (Genesee Scientific, El Cajon, CA, USA) with ITS1F and ITS4 primers [38]. The thermal cycling program included activation of the polymerase at 95˚C for 15 minutes followed by 30 cycles of denaturation (95˚C, 30 seconds), annealing (55˚C, 30 seconds), and extension (72˚C, 48 seconds), then ended with a final extension (72˚C, 7 minutes). Primers and dNTPs were eliminated using exonuclease I and shrimp alkaline phosphatase (New England BioLabs, Ipswich, MA, USA) and sent to the Brigham Young University DNA Sequencing Center (https://biology.byu.edu/dnasc) for Sanger sequencing. Sequences were trimmed to exclude quality scores below 10 in CodonCode Aligner (CodonCode Corporation, Centerville, MA, USA, v. 9.0.1). The average trimmed sequence length was approximately 530 bp. Sequences were then searched against the UNITE database (https://unite.ut.ee/index.php) using BLASTn. We used a 99% identity criterion to match to a species and 97% identity to match to a genus. Five isolates could not be identified to a species or genus in the database. We, therefore, constructed a phylogenetic tree to assist in identifying these taxa. All sequences were aligned in MAFFT (https://mafft.cbrc.jp/alignment/server/) [39,40] and cleaned in Mesquite v. 3.70 [41]. The output from Mesquite was used to construct a tree in IQTree (http://iqtree.cibiv.univie.ac.at/) [42], which was visualized in Fig-Tree v. 1.4.4. (http://tree.bio.ed.ac.uk/software/figtree/).

## Testing hypothesis 1: Psychrotolerance

Each of the 40 isolates was grown in a 2 x 2 factorial experiment with two incubation temperatures (5 and 17˚C) and two media (control and leaf litter). Our criterion for psychrotolerance was statistically significant growth at 5˚C. See below for greater detail. The warmer temperature (17˚C) represents the temperature during the growing season for *Q. gambelii*. The actual daily average temperature at Devil's Kitchen during the majority of the growing season (24 Jun 2019 to 24 Sep 2019) was 16.9˚C (SD, 6.2). The control medium contained starter glucose. The leaf litter medium contained starter glucose and leaf litter. There were 4 replicate 5 cm diameter dishes for each medium x temperature combination, 16 total dishes per isolate, for a total of 640 dishes. Each of the dishes was inoculated with the appropriate isolate using a 6 mm diameter piece of mycelium taken from the outer edge of the parent culture. Each liter of control medium contained 2.0 g glucose, 8.92 g agar, 0.46 g peptone, 1.0 ml of 300 g $CaCl_2$ $L^{-1}$,

10.0 ml of 30 g $KH_2PO_4$ $L^{-1}$, and 1.0 ml of a solution containing 5 g $MgSO_4$ $L^{-1}$, 3.7 g $FeSO_4$ $L^{-1}$, 1.4 g $MnSO_4$ $L^{-1}$ and 3.7 g $ZnSO_4$ $L^{-1}$. Each liter of leaf litter medium additionally contained 7.18 g of *Q. gambelii* leaf litter, ground to pass a 2 mm screen in a Cyclone Mill (Retsch USA, Verder Scientific, Newtown, PA, USA). All media components were added prior to autoclaving. The litter had been collected from the forest floor of the Slate Canyon site in the summer of 2020.

The experiment was carried out in sets of 4 to 6 isolates. There were 8 sets total. Biomass growth rates were determined after 49–53 d of growth, depending on the set. To determine fungal biomass, the agar within each dish was melted within a stainless-steel tea ball (0.6 mm mesh) in boiling water for 10 min., the freed mycelium was dried on the lab bench at room temperature for 24 hours and weighed. It is unlikely that any of the fungal biomass was lost through the stainless-steel mesh of the tea ball because each mycelium consisted of a solid fungal mat. The biomass growth rate was calculated using the following formula: $(DW_2 - DW_1)$ / elapsed time, for which we assumed $DW_1 = 0$. In reality, the mycelium dry weight for any replicate at the beginning of the experiment was not zero, but was less than 0.0001 g, the lower limit of our balance.

To test hypothesis 1, we defined psychrotolerance as significant saprotrophic capacity at 5˚C. We defined saprotrophic capacity as the difference between the biomass growth rate on leaf litter medium (containing starter glucose and leaf litter), and the biomass growth rate on control medium (containing starter glucose). At 5˚C, there were 4 replicates for each of the media, but leaf litter replicates were not paired with corresponding control replicates for the purpose of calculating differences in growth rate (saprotrophic capacity). Therefore, we calculated the error associated with the difference by bootstrapping. First, differences were determined for the 16 possible combinations of leaf medium (4 replicates) and control medium (4 replicates). Then, 4 of the 16 differences were randomly sampled 1000 times with replacement in order to bootstrap a frequency distribution of the difference, and the mean, standard deviation and 95% confidence intervals were determined in R [43]. If this difference was significantly greater than zero for isolates grown at 5˚C, according to the 95% confidence interval, we considered the isolate to be a psychrotolerant saprotroph. The R script for this procedure is given in the supporting information (S1 File).

We analyzed the variation in psychrotolerant saprotrophy (saprotrophic capacity at 5˚C) among isolates. Again, because leaf litter replicates were not paired with corresponding control replicates for the purpose of calculating differences in growth rate (= saprotrophic capacity), it was not possible to perform a standard analysis of variance of the difference in growth rate. Therefore, the analysis of variance was performed in R (see https://acetabulum.dk/) using the means and standard deviations of the differences in growth rate at 5˚C obtained from the bootstrapping R script to analyze the variation among isolates in psychrotolerant saprotrophy (saprotrophic capacity at 5˚C). To determine whether psychrotolerant saprotrophy differed among species, a conventional analysis of variance was performed with isolates as replicates within the four species comprising multiple isolates using Minitab v. 18 [44].

### Testing hypothesis 2: Temperature specialization

To test hypothesis 2, we performed a linear correlation of saprotrophic capacities of isolates at 17˚C and at 5˚C. The P and r values were obtained through SigmaPlot for Windows, Version 11.0 [45].

### Results

Based on the UNITE database, 35 of our 40 isolates were identified to the following taxa: *Apiognomonia errabunda* (Roberge ex Desm.) Höhn, *Cladosporium herbarum* (Pers.) Link Ex

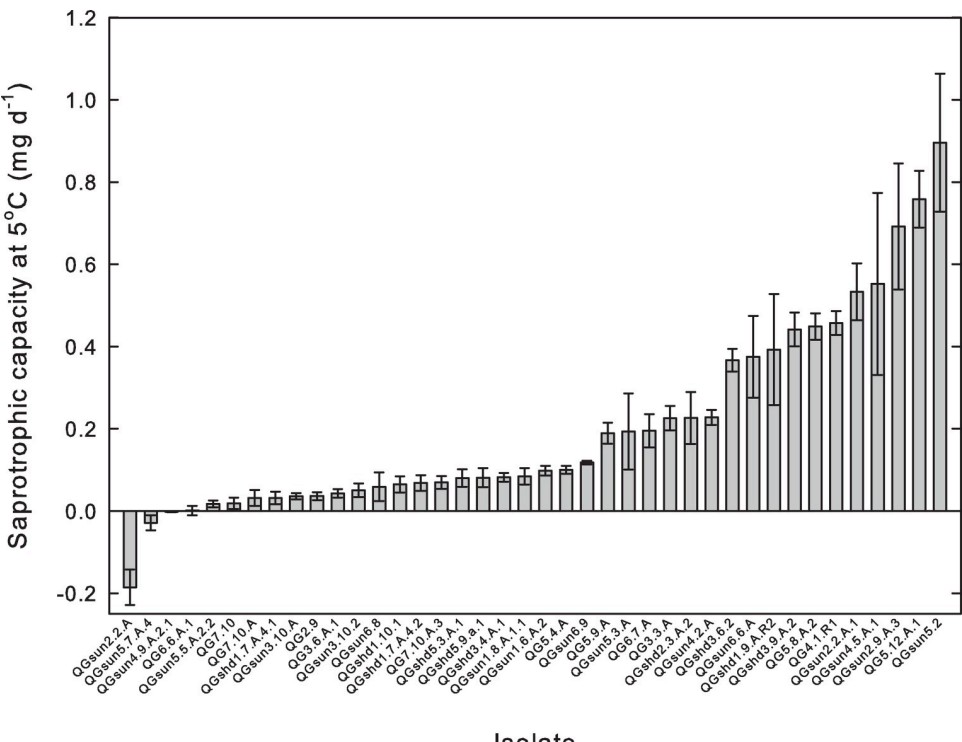

**Fig 1. Psychrotolerant saprotrophy (saprotrophic capacity at 5˚C) among the 40 isolates.** Error bars are 95% confidence intervals. The list of the fungal taxa for each of these isolates is given in the supporting information (S1 Table).

S. F. Gray, *Ophiognomonia setacea* (Pers.) Sogonov, *Ophiognomonia* sp., *Pyronema omphalodes* Bull. (Fuckel), *Venturia* sp., *Coniochaeta polymorpha* Z.U. Khan, J.P. Guarro & S. Ahmad, Antonie van Leeuwenhoek, *Cladosporium allicinum* (Fr.) Bensch, U. Braun & Crous, 2012, *Tricharina cretea* (Cooke) K.S. Thind & Waraitch, *Parafenestella* sp. The remaining 5 isolates were identified using the constructed phylogenetic tree: one isolate each in the Dothideales, Diaporthales, Coniochaetales, and two isolates in the Pezizomycetes.

Thirty-six of the 40 isolates had significant saprotrophic capacity at 5˚C or, in other words, were psychrotolerant saprotrophs (Fig 1). In contrast, 4 isolates (QGsun2.2.A, *Ophiognomonia setacea*; QGsun5.7.A.4, *Coniochaetales*.; QGsun4.9.A.2.1, *Pryonema omphalodes*; QG6.6.A.1, *Ophiognomonia sp.*) did not exhibit significant psychrotolerant saprotrophy.

There was significant variation in psychrotolerant saprotrophy among fungal isolates (Fig 1, Table 1). Isolate was a significant factor (p = 6.42e$^{-60}$), accounting for 95% of total variability in psychrotolerant saprotrophy, while the variability among replicates within an isolate accounted for only 5% (Table 1).

When we analyzed the four fungal species comprising more than a single isolate (Fig 2), species was a significant factor with respect to psychrotolerant saprotrophy (p = 0.007), with

**Table 1. ANOVA table for psychrotolerant saprotrophy (saprotrophic capacity at 5˚C) among the 40 isolates.**

| Source | df | SS | MS | F | P |
|---|---|---|---|---|---|
| Among isolates | 39 | 8.8673 | 0.22737 | 54.6571 | 6.42e$^{-60}$ |
| Within isolates | 120 | 0.4992 | 0.00416 | | |
| Total | 159 | 9.3664 | | | |

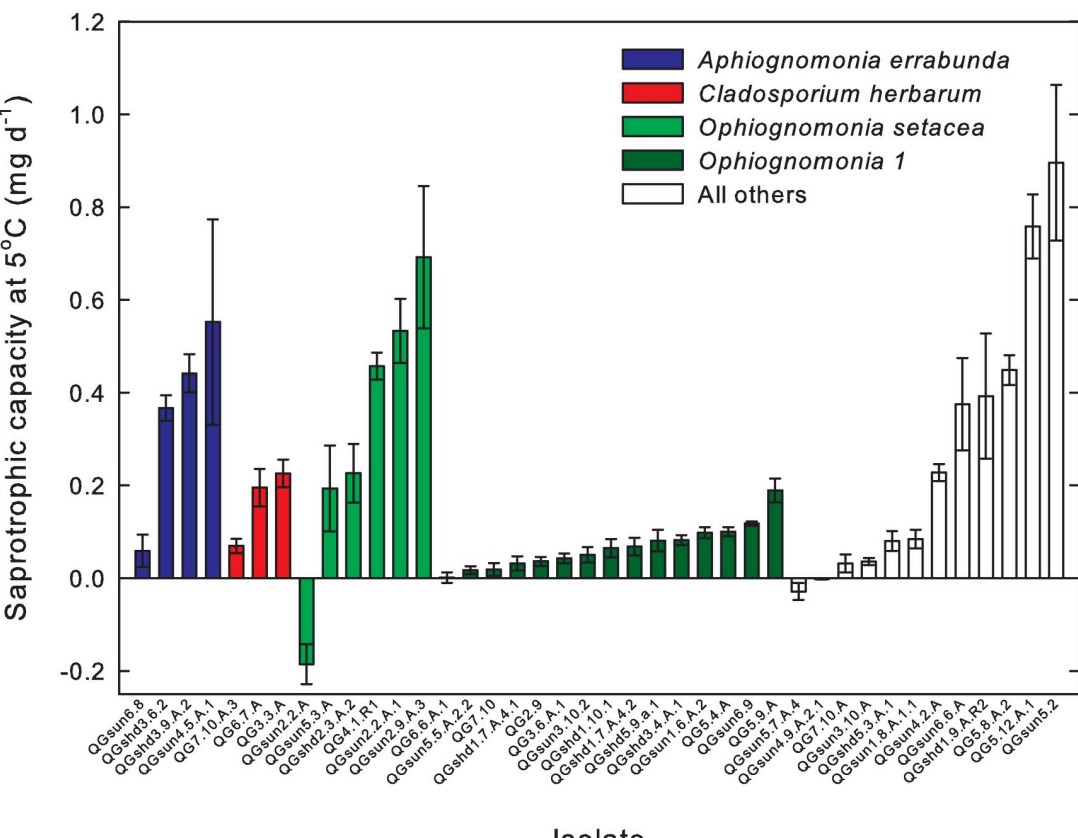

**Fig 2. Psychrotolerant saprotrophy (saprotrophic capacity at 5°C) among the 40 isolates, grouped by species.** The white bars represent taxa that were either species comprising a single isolate each or were not identified to species. Such taxa (white) were not included in the analysis of variance shown in Table 2. Error bars are 95% confidence intervals. The list of the fungal taxa for each of these isolates is given in the supporting information (S1 Table).

among species variation accounting for 39% of the total variability (Table 2). *Ophiognomonia sp.* had a saprotrophic psychrotolerance significantly lower than that of *Apiognomonia errabunda* and *Ophiognomonia setacea* (Table 3). However, within species variability was also large, accounting for 61% of total variability (Table 2).

The correlation between the saprotrophic capacity of isolates at 17°C and 5°C was positive and significant (p = 0.0003, r = 0.5434, Fig 3).

## Discussion

Because much of the microbially-mediated decomposition of *Q. gambelii* leaf litter must occur in the colder months of the year when moisture is available, we hypothesized that endophytic fungi colonizing *Q. gambelii* leaves were capable of saprotrophy at 5°C. Of the endophytic

**Table 2. ANOVA table for psychrotolerant saprotrophy (saprotrophic capacity at 5°C) among the four species each comprising multiple isolates.**

| Source | df | SS | MS | F | P |
|---|---|---|---|---|---|
| Among species | 3 | 0.4306 | 0.14353 | 5.2 | 0.007 |
| Within species | 24 | 0.6628 | 0.02762 | | |
| Total | 27 | 1.0933 | | | |

**Table 3. Means and standard errors of the means (SEM) for psychrotolerant saprotrophy (saprotrophic capacity at 5˚C) among the four species each comprising multiple isolates.** Different letters indicate a significant difference among species according to the Tukey Honestly Significant Difference Method.

| Species | n | Mean (SEM) |
|---|---|---|
| *Apiognomonia errabunda* | 4 | 0.3549 (0.0831) a |
| *Ophiognomonia setacea* | 6 | 0.3194 (0.0678) a |
| *Cladosporium herbarum* | 3 | 0.1637 (0.0959) ab |
| *Ophiognomonia sp.* | 15 | 0.0668 (0.0429) b |

fungi we isolated, saprotrophic psychrotolerance was common but not universal, occurring in all but 4 of the 40 tested isolates. Others have reported psychrotolerant growth of endophytic fungi on glucose-based media [34,46]. However, this constitutes the first report of psychrotolerant saprotrophy of endophytic fungi using leaf litter as the carbon source.

We also hypothesized that there was significant variation among species in saprotrophic psychrotolerance. While the species designated *Ophiognomonia sp.* had a significantly lower saprotrophic psychrotolerance than the other species, there was a surprising level of variation within species as was seen for *Aphiognomonia errabunda*, *Cladosporium herbarum* and *Ophiognomonia setacea*. This result was unexpected because members of a given species are assumed to occupy the same niche and, therefore, to be functionally similar [47]. We note, however, that the unexpectedly high degree of variation in psychrotolerance within a species was expressed under artificial experimental conditions, including controlled temperatures and

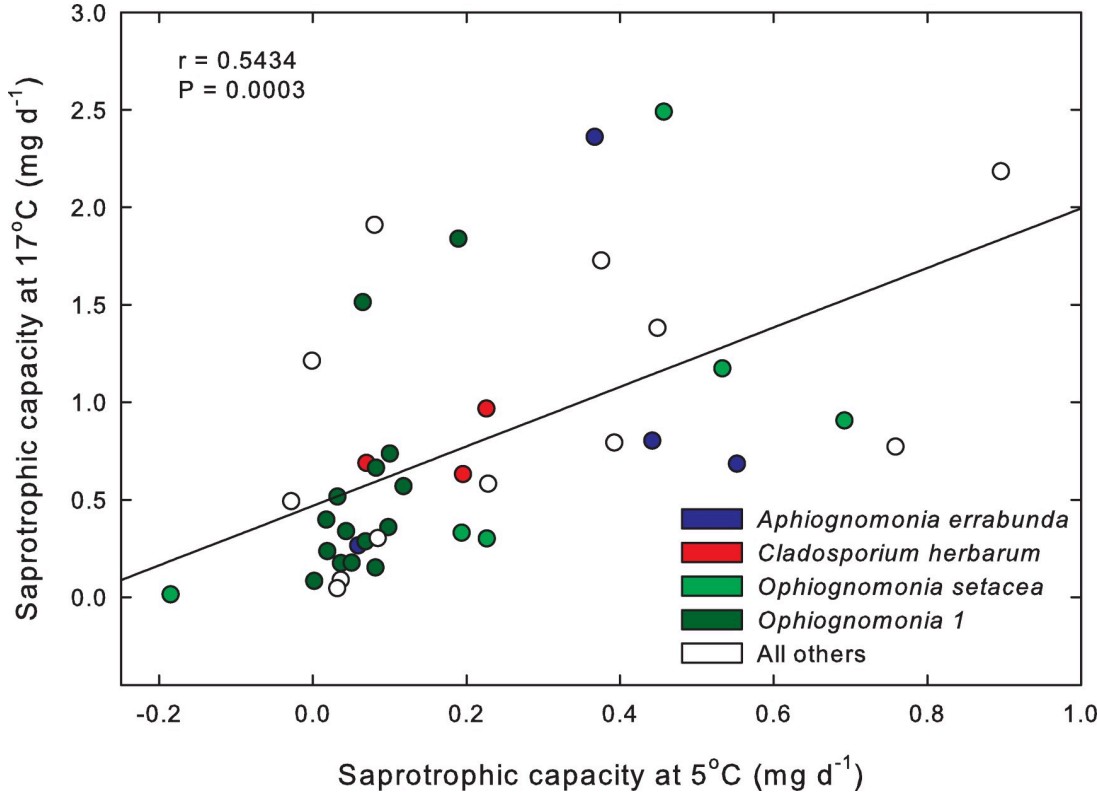

**Fig 3. Plot of saprotrophic capacity of the 40 isolates at 17˚C vs. 5˚C.** Blue is *Apiognomonia errabunda*, red is *Cladosporium herbarum*, light green is *Ophiognomonia setacea*, dark green is *Ophiognomonia sp.*, and white are taxa that were either not identified to species or species comprising a single isolate each.

non-limiting availabilities of water and mineral nutrients. Under field conditions, where cold or insufficiency of water or mineral nutrients could limit saprotrophic activity, the ability to express variation in psychrotolerance may be more limited.

Among the 40 fungal isolates, we found no evidence of a tradeoff in saprotrophic capacity at 17˚C with saprotrophic capacity at 5˚C. In fact, the significant correlation between saprotrophic capacity at 17˚C and at 5˚C was positive. Therefore, these fungi appear not to specialize with respect to temperature. This result was somewhat unexpected given the existence of performance tradeoffs at different temperatures for important physiological processes such as photosynthesis [32]. The capacity to obtain nutrition saprotrophically under a wide range of temperatures, from 5˚C to 17˚C, could potentially allow endophytic fungi to obtain resources from leaf litter both during the colder months of late autumn, winter, and early spring, when moisture is most abundant, as well as during the late summer monsoon when limited rain falls [2]. We have not assessed the ability of endophytic fungi to compete against non-endophytic fungi in litter, and it is possible that endophytic fungi are generally not highly competitive. However, we previously showed that endophytic fungi do at least persist in leaf litter of *Q. gambelii* for weeks following leaf senescence [21]. The persistence of endophytic fungi in decomposing litter is not particularly surprising given that when leaves senesce and enter the litter pool, endophytic fungi are already present, have no need to colonize from the environment, and thus have priority [25] over non-endophytic fungi in capturing resources from leaf litter. Indeed, others have found that endophytic fungi can exert priority over decomposer fungi in litter [28].

As with all laboratory studies, the ecological relevance of our results should be carefully weighed. This study was carried out under controlled temperatures (constant 17˚C vs. constant 5˚C), presumably without moisture or mineral nutrient limitation, and with every isolate grown separately, all conditions that are not likely to occur under field conditions. Because we do not know how these factors influence either saprotrophic capacity or psychrotolerance of a given isolate, the extent to which our results are relevant in the field is not clear. Nevertheless, our results suggest that a number of endophytic fungi of *Q. gambelii* leaves are potentially capable of saprotrophic growth under both cold and warm conditions and thus have the potential to influence litter decomposition and nutrient cycling in this arid ecosystem.

## Supporting information

**S1 Table. Isolate designation and species identity.**
(DOCX)

**S1 File. Bootstrapping R script.**
(DOCX)

**S2 File. Data set.**
(XLSX)

## Author Contributions

**Conceptualization:** Roger T. Koide.

**Data curation:** Emily Weatherhead, Emily Lorine Davis, Roger T. Koide.

**Formal analysis:** Emily Weatherhead, Emily Lorine Davis.

**Funding acquisition:** Roger T. Koide.

**Investigation:** Emily Weatherhead, Emily Lorine Davis.

**Methodology:** Roger T. Koide.

**Project administration:** Roger T. Koide.

**Resources:** Roger T. Koide.

**Supervision:** Roger T. Koide.

**Visualization:** Emily Weatherhead, Emily Lorine Davis.

**Writing – original draft:** Emily Weatherhead, Emily Lorine Davis.

**Writing – review & editing:** Roger T. Koide.

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
