## [Decision Letter · Decision Letter 0]

10 Jul 2022

PONE-D-22-12165Many foliar endophytic fungi of Quercus gambelii are capable of psychrotolerant saprotrophic growthPLOS ONE

Dear Dr. Koide,

Thank you for submitting your manuscript to PLOS ONE. After careful consideration, we feel that it has merit but does not fully meet PLOS ONE’s publication criteria as it currently stands. Therefore, we invite you to submit a revised version of the manuscript that addresses the points raised during the review process.

We look forward to receiving your revised manuscript.

Kind regards,

Raffaella Balestrini

Academic Editor

PLOS ONE

Journal Requirements:

“The authors acknowledge financial support from the Roger Sant Foundation and the Department of Biology, Brigham Young University.”

“The authors acknowledge financial support from the Roger Sant Foundation and the Department of Biology, Brigham Young University.”

Reviewers' comments:

Reviewer's Responses to Questions

**Comments to the Author**

1. Is the manuscript technically sound, and do the data support the conclusions?

Reviewer #1: Yes

Reviewer #2: Yes

2. Has the statistical analysis been performed appropriately and rigorously? 

Reviewer #1: Yes

Reviewer #2: Yes

3. Have the authors made all data underlying the findings in their manuscript fully available?

Reviewer #1: Yes

Reviewer #2: No

4. Is the manuscript presented in an intelligible fashion and written in standard English?

Reviewer #1: Yes

Reviewer #2: Yes

5. Review Comments to the Author

Reviewer #1: The paper is well written, well organized, and absolutely clear about how the authors conducted the research. The study could be of certain interest to the target of Plos One.

Nevertheless, the overall quality of the manuscript needs to be slightly improved before publishing in Plos One, since minor issues are present.

I suggest the authors to better explain the second hypothesis in the material and method section since just a few words were spent about the goal of this second part of the paper.

Moreover, it is important to improve the clarity and quality of the tables provided in the text deriving from ANOVA analyses; it is not specified the number of ways (one way or two way) as well as the meaning of the acronyms.

Furthermore, It would be useful to provide more details on the result of the statistical analysis reported in the tables within the text. Which variance gave greater significance? Between groups or within groups? This is important since It seems that these parameters differ a lot in terms of explained variance.

Minor issues

The figures contains a lot of data, but are not properly legible. Is it possible to rotate the figures? I understand that could negatively affect the figure quality, but It surely could improve from a legibility point of view.

Reviewer #2: I think the authors satisfactorily replied to the reviewer 1 which raise some issues that go beyond the aim of the work. Moreover, some comments are inappropriate (e.g., since mid-2000s direct PCR is successfully applied to fungal material without any problem about the sequence quality). The aim of the work is clear and the biological hypotheses which the work is based on are appropriate and fully met. Experimental planning and procedures are detailed and they support appropriately what the authors intended to demonstrate although some parts of the M&M section can be improved (see the below comments). However, it is not clear because you selected as “warm” temperature 17 °C that in many parts of the world is not so “warm”. I think this has to do with the climate where Q. gambelii grow. In this case the choice of this temperature level should be justified and an in-depth description of the study site should be reported (e.g., min and max temperatures of coldest and warmest months). The results about strain identification (ln 114-120) should be moved at the beginning of the result section and the phylogenetic tree generated to identify the unknown isolates should be shown. This part is a result and not a methodology and, in addition, this shift would compensate the gap of length between M&M and Result sections.

Accession numbers and site of collection of each isolate should be added in the supplementary Table 1

Specific comments

Ln 40, a brief description of the differences between psychrotolerant, psychrotrophic and psychrophilic should be provided to ensure that the reader understand why you chose the first term.

Ln 85, delete “2000”

Ln 88, how many sub-samples (disks) from each leaf?

Ln 92, “very few” is too subjective, “less than xx” would be better

Ln 92-96, I suppose the plates were inspected every day to isolate strains before R selected fungi (e.g. Penicillium, Trichoderma,…) form and release conidia

Ln 103, “ITS is considered a universal barcode for fungi” delete this sentence because superfluous

Ln 116, 119,…, “sp.” must not be italicized; What is “Ophiognomonia 1”? perhaps “Ophiognomonia sp.”?

Ln 132, for a total of 640 plates, is it correct?

Ln 133-136, it is not clear which is the size of the plates. I do not think that 200 ml of medium was added to each plate. Moreover the amount of each component of the medium should be reported as weight (g, mg,…) per L.

Ln 137, was the leaf litter autoclaved?

Ln 143, how did you separate mycelium biomass from residues of litter?

6. PLOS authors have the option to publish the peer review history of their article (what does this mean?). If published, this will include your full peer review and any attached files.

Reviewer #1: No

Reviewer #2: No

---

## [Author Response · Author response to Decision Letter 0]

29 Jul 2022

There is no need to make changes to the financial disclosure statement. Thank you.

Journal Requirements: 

I have made the prescribed changes to the manuscript.

This statement is now included in the methods section.

“The authors acknowledge financial support from the Roger Sant Foundation and the Department of Biology, Brigham Young University.”

“The authors acknowledge financial support from the Roger Sant Foundation and the Department of Biology, Brigham Young University.”

The funding statement is sufficient, and I have eliminated the acknowledgement.

There is no need to make changes to the funding statement. Thank you.

The data set is now provided as a supporting information file (S3_File).

Captions for the Supporting Information files have now been added to the manuscript.

The reference list has been updated. 

Reviewers' comments:

Reviewer's Responses to Questions 

Comments to the Author

1. Is the manuscript technically sound, and do the data support the conclusions?

Reviewer #1: Yes

Reviewer #2: Yes

2. Has the statistical analysis been performed appropriately and rigorously? 

Reviewer #1: Yes

Reviewer #2: Yes

3. Have the authors made all data underlying the findings in their manuscript fully available?

Reviewer #1: Yes

Reviewer #2: No. The dataset is now available as a supplemental file.

4. Is the manuscript presented in an intelligible fashion and written in standard English?

Reviewer #1: Yes

Reviewer #2: Yes

5. Review Comments to the Author

Reviewer #1: The paper is well written, well organized, and absolutely clear about how the authors conducted the research. The study could be of certain interest to the target of Plos One.

Nevertheless, the overall quality of the manuscript needs to be slightly improved before publishing in Plos One, since minor issues are present.

I suggest the authors to better explain the second hypothesis in the material and method section since just a few words were spent about the goal of this second part of the paper.

The second hypothesis is given in the introduction. It is a straightforward hypothesis about an ecological tradeoff, and other examples of such tradeoffs are given in the introduction for clarification. Therefore, there doesn’t seem to be a need to further explain this hypothesis in the materials and methods section, which is devoted only to explaining the methods used to test the hypothesis.

Moreover, it is important to improve the clarity and quality of the tables provided in the text deriving from ANOVA analyses; it is not specified the number of ways (one way or two way) as well as the meaning of the acronyms.

The structure of the tables indicates that both anovas were one-way analyses. I am not sure what the reviewer means as far “acronyms” are concerned. The “df”, “SS”, “MS”, “F” and “P” are straightforward abbreviations of well-known anova terms (degrees of freedom, sums of squares, mean squares, F statistic and P value). If a reader does not recognize these terms, they will not be able to decipher an anova table anyway. 

Furthermore, It would be useful to provide more details on the result of the statistical analysis reported in the tables within the text. Which variance gave greater significance? Between groups or within groups? This is important since It seems that these parameters differ a lot in terms of explained variance.

The variance is given by the sums of squares. The very reason we included the complete anova table was to allow the reader to make this assessment. Indeed, in the results section, we already make statements such as “with among species variation accounting for 39% of the total variability”. 

Minor issues

The figures contains a lot of data, but are not properly legible. Is it possible to rotate the figures? I understand that could negatively affect the figure quality, but It surely could improve from a legibility point of view.

Yes, the figures could be rotated, but I do not see how that would increase how easily the figures are understood. Species names, some of which are very long, are not given in these figures to keep the figure as simple and easily interpreted as possible. S2 Table lists the species name for each of the isolates. 

Reviewer #2: I think the authors satisfactorily replied to the reviewer 1 which raise some issues that go beyond the aim of the work. Moreover, some comments are inappropriate (e.g., since mid-2000s direct PCR is successfully applied to fungal material without any problem about the sequence quality). The aim of the work is clear and the biological hypotheses which the work is based on are appropriate and fully met. Experimental planning and procedures are detailed and they support appropriately what the authors intended to demonstrate although some parts of the M&M section can be improved (see the below comments). 

Thank you for that confirmation.

However, it is not clear because you selected as “warm” temperature 17 °C that in many parts of the world is not so “warm”. I think this has to do with the climate where Q. gambelii grow. In this case the choice of this temperature level should be justified and an in-depth description of the study site should be reported (e.g., min and max temperatures of coldest and warmest months). 

We now indicate in the materials and methods section that “air temperatures were monitored at the Devil’s Kitchen site from 24 Jun 2019 to 24 Sep 2019 using a temperature data logger (HOBO UA-001-64, Onset Computer Corporation, Bourne, MA, USA), set to log hourly.” Furthermore, in the section “Testing hypothesis 1: psychrotolerance”, we added “Each of the 40 isolates was grown in a 2 x 2 factorial experiment with two incubation temperatures (5 and 17 ºC) and two media (control and leaf litter). Our criterion for psychrotolerance was statistically significant growth at 5 ºC. The warmer temperature (17 ºC) represents the temperature during the growing season for Q. gambelii. The actual daily average temperature at Devil’s Kitchen during the majority of the growing season (24 Jun 2019 to 24 Sep 2019) was 16.9 �C (SD, 6.2).”

The results about strain identification (ln 114-120) should be moved at the beginning of the result section and the phylogenetic tree generated to identify the unknown isolates should be shown. This part is a result and not a methodology and, in addition, this shift would compensate the gap of length between M&M and Result sections.

As suggested, the identity of the isolates has been moved to the results section. However, we consider the generation of the phylogenetic tree to be part of the methods section because it was used to identify five of the unknown isolates. Therefore, the section title for this part of the materials and methods has been changed to “Sequencing and assigning taxonomy of fungal isolates”. Because the tree was used as a method and was not a result, we do not feel it should be included as a result.

Accession numbers and site of collection of each isolate should be added in the supplementary Table 1

The UNITE ID numbers and collection sites have been added to S2 Table.

Specific comments

Ln 40, a brief description of the differences between psychrotolerant, psychrotrophic and psychrophilic should be provided to ensure that the reader understand why you chose the first term.

Unfortunately, these terms have been used interchangeably depending on the author. Therefore, any of the terms could have been used. We chose psychrotolerant because the condition of a psychrotolerant organism is easily described as “psychrotolerance”, whereas the conditions describing a psychrotrophic or psychrophilic organism are awkward.

Ln 85, delete “2000”

Done.

Ln 88, how many sub-samples (disks) from each leaf?

We now clarify that “Each surface-sterilized leaf was subsampled once…”

Ln 92, “very few” is too subjective, “less than xx” would be better

The statement now reads “After 3 weeks, fewer than 10 total fungal colonies grew out from the entire collection of leaf disks.”

Ln 92-96, I suppose the plates were inspected every day to isolate strains before R selected fungi (e.g. Penicillium, Trichoderma,…) form and release conidia

Yes.

Ln 103, “ITS is considered a universal barcode for fungi” delete this sentence because superfluous

Deleted

Ln 116, 119,…, “sp.” must not be italicized; What is “Ophiognomonia 1”? perhaps “Ophiognomonia sp.”?

These have been corrected throughout the manuscript.

Ln 132, for a total of 640 plates, is it correct?

Yes, this fact has been added.

Ln 133-136, it is not clear which is the size of the plates. 

We now specify that the plates were 5 cm diameter.

I do not think that 200 ml of medium was added to each plate. Moreover the amount of each component of the medium should be reported as weight (g, mg,…) per L.

The components of the media are now given per liter.

Ln 137, was the leaf litter autoclaved?

Yes, the leaf litter was added to the medium prior to autoclaving. We now write “All media components were added prior to autoclaving.”

Ln 143, how did you separate mycelium biomass from residues of litter?

Surprisingly, the vast majority of the ground leaf litter was easily separated from the mycelium during the boiling process. 

Do you want your identity to be public for this peer review? For information about this choice, including consent withdrawal, please see our Privacy Policy.

Reviewer #1: No

Reviewer #2: No

---

## [Decision Letter · Decision Letter 1]

26 Sep 2022

Many foliar endophytic fungi of Quercus gambelii are capable of psychrotolerant saprotrophic growth

PONE-D-22-12165R1

Dear Dr. Koide,

We’re pleased to inform you that your manuscript has been judged scientifically suitable for publication and will be formally accepted for publication once it meets all outstanding technical requirements.

Kind regards,

Raffaella Balestrini

Academic Editor

PLOS ONE

Additional Editor Comments (optional):

Reviewers' comments:

Reviewer's Responses to Questions

**Comments to the Author**

1. If the authors have adequately addressed your comments raised in a previous round of review and you feel that this manuscript is now acceptable for publication, you may indicate that here to bypass the “Comments to the Author” section, enter your conflict of interest statement in the “Confidential to Editor” section, and submit your "Accept" recommendation.

Reviewer #1: All comments have been addressed

Reviewer #2: All comments have been addressed

2. Is the manuscript technically sound, and do the data support the conclusions?

Reviewer #1: Yes

Reviewer #2: Yes

3. Has the statistical analysis been performed appropriately and rigorously? 

Reviewer #1: Yes

Reviewer #2: Yes

4. Have the authors made all data underlying the findings in their manuscript fully available?

Reviewer #1: Yes

Reviewer #2: Yes

5. Is the manuscript presented in an intelligible fashion and written in standard English?

Reviewer #1: Yes

Reviewer #2: Yes

6. Review Comments to the Author

Reviewer #1: After a careful and considered review of the content of this paper by the authors, the article was now found to be available for publication.

Reviewer #2: The revised version of the manuscript can be now published on PlosOne. All comments have been satisfied appropiately.

7. PLOS authors have the option to publish the peer review history of their article (what does this mean?). If published, this will include your full peer review and any attached files.

Reviewer #1: No

Reviewer #2: No

---

## [Editor Report · Acceptance letter]

3 Oct 2022

PONE-D-22-12165R1 

Many foliar endophytic fungi of *Quercus gambelii* are capable of psychrotolerant saprotrophic growth 

Dear Dr. Koide:

I'm pleased to inform you that your manuscript has been deemed suitable for publication in PLOS ONE. Congratulations! Your manuscript is now with our production department. 

Kind regards, 

on behalf of

Dr Raffaella Balestrini 

Academic Editor

PLOS ONE